# Feature Extraction and Reconstruction by Using 2D-VMD Based on Carrier-Free UWB Radar Application in Human Motion Recognition

**DOI:** 10.3390/s19091962

**Published:** 2019-04-26

**Authors:** Liubing Jiang, Xiaolong Zhou, Li Che, Shuwei Rong, Hexin Wen

**Affiliations:** 1School of Computer and Information Security, Guilin University of Electronic Technology, Guilin 541004, China; jlrq1@163.com; 2Key Laboratory of Wireless Broadband Communication and Signal Processing in Guangxi, Guilin University of Electronic Technology, Guilin 541004, China; 18589949895@163.com (S.R.); winhexin@foxmail.com (H.W.); 3School of Information and Communication, Guilin University of Electronic Technology, Guilin 541004, China

**Keywords:** carrier-free UWB Radar, human motion recognition, 2D-VMD algorithm, feature extraction, reconstruction

## Abstract

As the size of the radar hardware platform becomes smaller and smaller, the cost becomes lower and lower. The application of indoor radar-based human motion recognition has become a reality, which can be realized in a low-cost device with simple architecture. Compared with narrow-band radar (such as continuous wave radar, etc.), the human motion echo signal of the carrier-free ultra-wideband (UWB) radar contains more abundant characteristic information of human motion, which is helpful for identifying different types of human motion. In this paper, a novel feature extraction method by two-dimensional variational mode decomposition (2D-VMD) algorithm is proposed. And it is used for extracting the primary features of human motion. The 2D-VMD algorithm is an adaptive non-recursive multiscale decomposition method for nonlinear and nonstationary signals. Firstly, the original 2D radar echo signals are decomposed by the 2D-VMD algorithm to capture several 2D intrinsic mode function (BIMFs) which represent different groups of central frequency components of a certain type of human motion. Secondly, original echo signals are reconstructed according to the several BIMFs, which not only have a certain inhibitory effect on the clutter in the echo signal, but can also further demonstrate that the BIMFs obtained by the 2D-VMD algorithm can represent the original 2D echo signal well. Finally, based on the measured ten different types of UWB radar human motion 2D echo analysis signals, the characteristics of these different types of human motion are extracted and the original echo signal are reconstructed. Then, the three indicators of the *PCC*, *UQI*, and *PSNR* between the original echo signals and extraction/reconstruction 2D signals are analyzed, which illustrate the effectiveness of 2D-VMD algorithm to extract feature of human motion 2D echo signals of the carrier-free UWB radar. Experimental results show that BIMFs by 2D-VMD algorithm can well represent the echo signal characteristics of this type of human motion, which is a very effective tool for human motion radar echo signal feature extraction.

## 1. Introduction

Due to the defense of national defense borders, pedestrian identification of automobile safety, and fall detection for assisting the elderly, research about human motion recognition based on radar attract interests of many researchers [1]. The research goal is to identify and alert human action behaviors based on the prior behavioral information, it plays a certain predictive role, mainly for hospital patient monitoring, urban struggles, and search & rescue operations. With the smaller and smaller cost of the software defined and wireless radar platform, radar-based human motion recognition research can distinguish different types of motions that are similar to each other, and the application of indoor radar human motion recognition has become a reality [2].

Most researches based on radar human motion recognition focus on extracting important and reliable features by collecting different types of human motion echo signal data and then classifying and identifying them. Thayaparan, T., Abrol, S., and Riseborough, E. [3,4] of the Canadian Defense Research and Development Center used the combined wavelet and time–frequency method to extract the micro-Doppler characteristics of human motion based on the radar system, and obtained the basic parameters of human motion. S.S. Ram of the University of Texas at Austin used the time–frequency rearrangement method to analyze the pedestrian echo micro-Doppler [5,6]. Its high resolution makes it possible to distinguish the nine parts of the body from the time–frequency image. However, the STFT method can only distinguish the micro-Doppler component of the body parts of three people. Bryan, J.D., Kwon, J. and Lee N. Studied the feasibility of using UWB radar to classify different human activities. UWB radar was used to collect eight different types of activities of eight different human activity, including walking, running, rotating, punching, jumping, transitioning between standing and sitting, crawling, and standing still. Then, The Pearson correlation coefficient (PCA) method was used to capture the feature in UWB radar echo signal. The feature set includes PCA coefficient, PCA mean and variance, and FFT transform result of PCA and target velocity. Finally, the classification accuracy is more than 85% [7]. Most of the existing researches on radar human motion recognition are based on frequency modulated continuous wave radar, which transmits and receives echo signals by the way of modulation and demodulation. However, our paper is aimed at carrier-free ultra-wideband radar in the form of a pulse wave. The echo signal of the radar-based detecting human motion is a nonlinear nonstationary signal, so some time–frequency analysis methods, (such as Fast Fourier Transformation (FFT) method, Short-time Fourier Transform (STFT) method, etc.) which should trade-off between time and frequency resolution lead to the poor performance of feature extraction in human motion echo signals.

Recently, various spectral decomposition techniques have been developed and applied, such as the short-time Fourier transform (STFT) [8], continuous wavelet transform [9], generalized S transformation [10], Wigner Ville distribution [11], and matching pursuit decomposition [12]. A kind of self-adaptive signal processing algorithm is empirical mode decomposition (EMD) [13,14], originally proposed by Huang et al. This is a data-driven and adaptive algorithm depends on the local characteristics of data in time domain. EMD decomposes the nonlinear and nonstationary signal into a series of intrinsic mode functions (IMFs) with definite physical meanings and instantaneous features [13]. It has a good time–frequency resolution and can be applied in many scientific fields. Zhao, G. and Lin, Q. proposed an EMD-based approach to UWB radar for sense-through-foliage target detection [15]. Wang, Y. and Wu, X. combined both EMD and Smoothed Pseudo Wigner–Vill Distribution (SPWVD) techniques for the micro-Doppler analysis [16]. However, there are some apparent limitations of EMD-based methods [17], such as the sensitivity to noise, the lack of mathematical theory, the problems of mode-mixing, and end effects.

In order to avoid the limitations to the disadvantage of EMD-based method, a newly developed mode decomposition method called variational mode decomposition (VMD) has been proposed by Dragomiretskiy and Zosso [18]. As a non-recursive decomposition method, VMD can decompose a multicomponent signal into a finite number of bandlimited IMFs simultaneously [19,20]. Different to the recursive screening mode of the EMD, the VMD algorithm not only has a solid theoretical foundation, but is also less susceptible to noise than the EMD-based decomposition method because Wiener filtering is embedded to update the mode directly in the Fourier domain, and controls the convergence condition to effectively eliminate mode mixing with a high noise immunity [21]. VMD in 1D is essentially based on well-established concepts such as the 1D Hilbert transform and the analytic signal, heterodyne demodulation, and Wiener filtering. The goal of 1D-VMD is to decompose an input signal into a discrete number of sub-signals (modes), where each mode has limited bandwidth in spectral domain. In other words, one requires each mode $\mu_{k}$: R→R to be mostly compact (spectrally concentrated) around a center pulsation, which is to be determined along with the decomposition [22]. Just like EMD was originally proposed for a 1-D signal and was extended to bidimensional signals by Huang [23], Dragomiretskiy and Zosso extended the VMD method to two-dimensional VMD (2D-VMD), which can adaptively decompose an image/2D analytic signal into a few different modes of separate spectral bands which have specific directional and oscillatory characteristics [24]. 2D-VMD leads to band-limited intrinsic mode functions (BLIMFs, 2D-BLIMFs for the bi-dimensional case). Each mode has a limited spectral bandwidth: the first BLIMF contains the highest frequencies and the last one contains the lowest frequencies. More precisely, VMD/2D-VMD concurrently consists of searching a specific number of modes and their respective center frequencies used to reproduce the original signal/image either exactly or in least-square sense. In the one-dimensional case (1D signal), VMD decomposes the signal into K discrete modes (BLIMFs) [25]. Each mode has a limited spectral bandwidth [26]: each mode $\mu_{k}$ is compact around a center pulsation determined during the decomposition. In this paper, inspired by all of the aforementioned papers, a novel method of feature extraction and reconstruction using the 2D-VMD algorithm application in human motion recognition based on carrier-free UWB radar is proposed.

In this article, a new novel method for feature extraction and reconstruction using 2D-VMD algorithm application in human motion recognition by carrier-free UWB radar is proposed. The VMD algorithm is extended to the 2D-VMD algorithm for 2D analysis signal, which has a specific advantage for feature extraction and reconstruction. Firstly, the 2D-VMD algorithm is used to decompose the human motion 2D UWB radar echo signals into series BIMFs and collect these BIMFs as the feature of human motion. Then, the reconstructed signal acquired by reconstructing useful BIMFs. Compared reconstructed signal with original 2D UWB radar echo signal, our paper proves the effectiveness of the 2D-VMD algorithm in extracting human motion features. Finally, compare the *PCC* between the original echo signal and the BIMFs (the components of mode feature extracted by the 2D-VMD algorithm), which shows that the extracted features can represent certain information of the original echo signals, and compare the *PCC*, *UQI*, and *PSNR* values of the reconstructed echo signals and originals further illustrate that the 2D-VMD algorithm is very useful in feature extraction of human motion echo signals by the carrier-free UWB radar.

The outline of the article is as follows. Section 2 introduces the theoretical basis of two-dimensional variational mode decomposition (2D-VMD) and the derivation of 2D analysis signals in 2D-VMD algorithm. The feature extraction and reconstruction model using 2D-VMD application in human motion recognition by carrier-free UWB radar is proposed in Section 3. The experimental results proposed in this paper for features extracting and reconstructing echo signals are elaborated in Section 4. The experimental results proposed in this paper for features extracting and reconstructing echo signals are elaborated in Section 5. The last section is the summary and outlook of this article.

## 2. Two-Dimensional Variational Mode Decomposition (2D-VMD) Algorithm

Dragomiretskiy and Zosso extended the VMD method to two-dimensional VMD(2D-VMD), which can adaptively decompose a 2D analysis signal into several different modes of separate spectral bands which have specific directional and oscillatory characteristics [24]. 2D-VMD leads to band-limited intrinsic mode functions (BLIMFs, 2D-BLIMFs for the bi-dimensional case). Each mode has a limited spectral bandwidth: the first BLIMF contains the highest frequencies and the last one contains the lowest frequencies. More precisely, VMD/2D-VMD consists of searching a specific number of modes and their respective center frequencies used to reproduce the original signal/image either exactly or in least-square sense. The modes are extracted concurrently. In the one-dimensional case (1D signal), VMD decomposes the signal into K discrete modes (BLIMFs). Each mode has a limited spectral bandwidth [18]: each mode is compact around a center pulsation determined during the decomposition.

### 2.1. Two-Dimensional Variational Modal Function Model

The core of the 2D-VMD algorithm is used to decompose a 2D analysis signal to be detected into a number of 2D intrinsic mode components ${\hat{u}}_{k}\left(\vec{x}\right)$, which can be calculated by the following steps.(1)The 2D signal ${\hat{u}}_{AS,k}(\vec{\omega })$ is subjected to a 2D-Hilbert transform to obtain an analytical signal of 2D signal ${\hat{u}}_{AS,k}(\vec{\omega })$, thereby calculating a signal-sided 2D spectrum of ${\hat{u}}_{AS,k}(\vec{\omega })$ [27,28].
(1)$${\hat{u}}_{AS,k}(\vec{x})={\hat{u}}_{k}(\vec{x})**\left(\delta \left(\langle \vec{x},{\vec{\omega }}_{k}\rangle \right)+\frac{j}{\pi \langle \vec{x},{\vec{\omega }}_{k}\rangle }\right)\delta \left(\langle \vec{x},{\vec{\omega }}_{k,⊥}\rangle \right)$$
where δ(t) is the unit impulse function; j is the imaginary unit; ** is the 2D convolution.(2)The parsed signal of the intrinsic mode component ${\hat{u}}_{AS,k}(\vec{\omega })$ is estimated for its center frequency $e^{-j\langle \vec{\omega },{\vec{\omega }}_{k}\rangle }$, and then the spectrum of each ${\hat{u}}_{AS,k}(\vec{\omega })$ is modulated onto its corresponding frequency baseband.
(2)$$\left[{\hat{u}}_{k}(\vec{x})*\left(\delta \left(\langle \vec{x},{\vec{\omega }}_{k}\rangle \right)+\frac{j}{\pi \langle \vec{x},{\vec{\omega }}_{k}\rangle }\right)\delta \left(\langle \vec{x},{\vec{\omega }}_{k,⊥}\rangle \right)\right]e^{-j\langle \vec{x},{\vec{\omega }}_{k}\rangle }$$(3)Finally, calculate the square of the demodulated signal gradient $L_{2}$ norm in Equation (2).
(3)$${\left\|\nabla \left[{\hat{u}}_{k}(\vec{x})*\left(\delta \left(\langle \vec{x},{\vec{\omega }}_{k}\rangle \right)+\frac{j}{\pi \langle \vec{x},{\vec{\omega }}_{k}\rangle }\right)\delta \left(\langle \vec{x},{\vec{\omega }}_{k,⊥}\rangle \right)\right]e^{-j\langle x,{\vec{\omega }}_{k}\rangle }\right\|}_{2}^{2}$$
where ∇ represents the second derivative.

The variational constraint model of the 2D analysis signal obtained by the above Equations (1)–(3).
(4)$$\begin{array}{l} \underset{\left\{{\vec{u}}_{k},{\vec{\omega }}_{k}\right\}}{\min}\left\{\sum_{k=1}^{K}{\left\|\nabla \left[{\hat{u}}_{k}(\vec{x})*\left(\delta \left(\langle \vec{x},{\vec{\omega }}_{k}\rangle \right)+\frac{j}{\pi \langle \vec{x},{\vec{\omega }}_{k}\rangle }\right)\delta \left(\langle \vec{x},{\vec{\omega }}_{k,⊥}\rangle \right)\right]e^{-j\langle \vec{x},{\vec{\omega }}_{k}\rangle }\right\|}_{2}^{2}\right\}\\ s.t.\sum_{k=1}^{K}{\hat{u}}_{k}(x)=f \end{array}$$
where ${\hat{u}}_{k}=\left\{u_{1},u_{2},\ldots ,u_{k}\right\}$ represents the set of each modes after 2D mode decomposition and $\omega_{k}=\left\{\omega_{1},\omega_{2},\ldots ,\omega_{k}\right\}$ represents the set of center frequencies corresponding to each mode after the 2D variational mode decomposition.

The above 2D-VMD constraint model is actually a two-dimensional (2D) constrained optimal solution problem. In order to solve the above optimization problem, this paper uses Lagrange multiplication with the augmented Lagrange matrix function L.

### 2.2. Solution of Two-Dimensional Variational Constraint Model

The two-dimensional variational mode decomposition (2D-VMD) constraint model in Section 2.2. It is actually an optimal model of two-dimensional constraints. In this section, the augmented Lagrange matrix function L by using Lagrange number multiplication is introduced. it can be obtained by Equation (5).
(5)$$\begin{array}{c} L\left(\left\{u_{k}\right\},\left\{\omega_{k}\right\},\lambda \right)=\alpha \sum_{k=1}^{K}{\left\|\nabla \left[{\hat{u}}_{k}(\vec{x})*\left(\delta \left(\langle \vec{x},{\vec{\omega }}_{k}\rangle \right)+\frac{j}{\pi \langle \vec{x},{\vec{\omega }}_{k}\rangle }\right)\delta \left(\langle \vec{x},{\vec{\omega }}_{k,⊥}\rangle \right)\right]e^{-j\langle \vec{x},{\vec{\omega }}_{k}\rangle }\right\|}_{2}^{2}\\ +{\left\|f-\sum_{k=1}^{K}u_{k}(x)\right\|}_{2}^{2}+\langle \lambda (x),f-\sum_{k=1}^{K}u_{k}(x)\rangle \end{array}$$
where α represents the penalty factor parameter and λ is the Lagrange multiplier.

The alternating direction multiplier algorithm is used to solve the optimal solution of the augmented Lagrange function L of the upper variational model.

The solution of two-dimensional variational constraint model also conforms to the form of wiener filter. The specific expression is as follows
(6)$${\hat{u}}_{k}^{n+1}\left(\vec{\omega }\right)=\frac{\hat{f}\left(\vec{\omega }\right)-\sum_{k=1}^{K}u_{k}(\omega )+\frac{\lambda (t)}{2}}{1+2\alpha {\left(\omega -\omega_{k}\right)}^{2}}$$

Similarly, the solution to update center frequency $\omega_{k}^{n+1}$ is
(7)$${\vec{\omega }}_{k}^{n+1}=\frac{\int_{\Omega_{k}}\vec{\omega }{\left|{\hat{u}}_{k}\left(\vec{\omega }\right)\right|}^{2}d\vec{\omega }}{\int_{\Omega_{k}}{\left|{\hat{u}}_{k}\left(\vec{\omega }\right)\right|}^{2}d\vec{\omega }}$$
where the iteration step criterion is Equation (8).
(8)$$\sum_{k=1}^{K}\frac{{\left\|{\hat{u}}_{k}^{n+1}-{\hat{u}}_{k}^{n}\right\|}_{2}^{2}}{{\left\|{\hat{u}}_{k}^{n}\right\|}_{2}^{2}}<\varepsilon$$

The 2D-VMD algorithm simply updates each intrinsic mode component directly in frequency until the iteration stop condition is satisfied, and then output each intrinsic mode component.

The 2D-VMD algorithm is a result obtained by continuously updating the 2D analysis signal in the frequency and then performing inverse Fourier transform. The 2D-VMD algorithm is shown in the Algorithm 1 and the specific process of the 2D-VMD algorithm is described as follows:

(a) Initialize $\left\{{\hat{u}}_{k}^{1}\right\}$, $\left\{{\hat{\omega }}_{k}^{1}\right\}$, $\left\{{\hat{\lambda }}^{1}\right\}$ and n.

(b) Update $u_{k}$ and $\omega_{k}$ in frequency according to the above formula.

(c) Update λ, where
(9)$${\hat{\lambda }}^{n+1}\left(\omega \right)={\hat{\lambda }}^{n}\left(\omega \right)+\tau \left(\hat{f}\left(\omega \right)-\sum_{k=1}^{K}{\hat{u}}_{k}^{n+1}\left(\omega \right)\right)$$

(d) Until $\sum_{k=1}^{K}\frac{{\left\|{\hat{u}}_{k}^{n+1}-{\hat{u}}_{k}^{n}\right\|}_{2}^{2}}{{\left\|{\hat{u}}_{k}^{n}\right\|}_{2}^{2}}<\varepsilon$, stop iteration.


**Algorithm 1: 2D-VMD**
Input: signal f(x), number of modes k, parameters αk,τ,ε.Output: modes uk(x), center frequencies ωk.Initialize {ωk0},{u^k0}←0,λ^0←0,n←0repeatn←n+1for k=1:K doCreate 2D mask for analytic signal Fourier multiplier: ℋkt+1(ω)←1+sgn(〈ωkt, ω〉)Update u^AS,k:
(10)u^AS,kt+1(ω)← ℋkt+1(ω)[f^(ω) − ∑i<ku^it+1(ω) − ∑i>ku^it (ω) + λ^t(ω)2 1 +2αk|ω −ωkt|2 ]Update ωk:(11)ωkt+1← ∫ℝ2ω|u^AS,kt+1(ω)|2dω∫ℝ2|u^AS,kt+1(ω)|2dωRetrieve uk: (12)ukt+1(x)← ℛ(ℱ−1{u^AS,kt+1(ω)})end forDual ascent (optional):
(13)λ^t+1(ω)←λ^t(ω)+τ(f^(ω)− ∑ku^kt+1(ω))Until convergence: (14)∑k‖u^kt+1 − u^kt‖ 22‖u^kt‖22 <ϵ

## 3. Feature Extraction and Reconstruction Model of Human Motion 2D Echo Signal Based on 2D-VMD by Carrier-Free UWB Radar

The signal transmitted by the carrier-free UWB radar is a carrier-free pulse signal with a pulse width of nanoseconds, and its spectral components extend from direct current (DC) to Gigabit. Therefore, the carrier-free UWB radar echo signal contains rich target information, which is beneficial to identification of the target. The carrier-free UWB radar has unparalleled unique advantages in target detection and recognition, shelter projection, anti-interference, and anti-reconnaissance. It can not only make up for the shortage of the traditional radar blind zone and low precision, but also has high distance resolution rate, small close distance blind zone, high target recognition rate, weak multipath interference, and so on [29]. As UWB radar transmits a low powered pulse, it can provide range information of the target without considering limits of frequency allocation. Furthermore, it has short pulse duration such that it can contain information about a human motion. The UWB radar also can penetrate the obstacles to identify human, especially in the complex indoor environment. Compared with narrow-band radar (such as continuous wave (CW) radar), the echo signals of human motion with carrier-free UWB radar contain rich human features and motion feature information, which is helpful for identifying different types of human motion.

In this paper, experimental data acquisition based on ground penetrating radar with carrier-free UWB radar system. The echo signal of the device with carrier-free UWB radar is a 2D analysis signal, which is formed by superimposing a number of radar echo signals reflected from different parts of the human. Since the signal of the carrier-free UWB radar does not contain carrier frequency information the energy is concentrated in a very narrow waveform, and the correlation between the transmitted signal and the echo signal is weak. Therefore, a human motion echo signal decomposition method based on the 2D-VMD algorithm to extract the characteristic of this type of human motion echo signal is proposed that utilizes several BIMFs components obtained by the 2D-VMD algorithm [30,31,32]; it decomposes the original radar echo signals to reconstruct original echo signal. The 2D-VMD algorithm is a non-recursive variable-scale composition iterative search variational model that decomposes the 2D human motion echo signal of the carrier-free UWB radar into several BIMFs mode components, which represent the primary feature of this type of human motion. Then, by means of reconstructing the echo signals using these BIMFs mode components, it is further illustrated that the BIMFs mode components obtained by the 2D-VMD algorithm as the main feature of this type human motion is a very effective and useful method.

Our paper proposes a feature extraction and reconstruction model framework for 2D echo signals of the carrier-free UWB radar based on 2D-VMD algorithm. The model proposed in this paper is mainly composed of three parts: (i) 2D-VMD algorithm, (ii) feature extraction, and (iii) reconstruction. The steps of the proposed model are in detail listed as follows and diagrammed in Figure 1.Step1:Acquire ten different types of human motion 2D echo signals using SIR-20 carrier-free UWB radar.Step2:Decompose the 2D echo signals of each type of human motion using 2D-VMD algorithm to extract 2D mode components of n BIMFs.Step 2.1:Load the 2D human motion echo analysis signal data of the carrier-free UWB radar.Step 2.2:Initialize $\left\{\omega_{k}^{0}\right\},\left\{{\hat{u}}_{k}^{0}\right\}\leftarrow 0,{\hat{\lambda }}^{0}\leftarrow 0,n\leftarrow 0$;Step 2.3:Update ${\hat{u}}_{AS,k}$, with Equation (10);Step 2.4:Update $\omega_{k}$, with Equation (11);Step 2.5:Update retrieve $u_{k}$, with Equation (12);Step 2.6:Update λ using Equation (13);Step 2.7:Step 2.7: Determine whether to converge using $\sum_{k}\frac{{\left\|{\hat{u}}_{k}^{t+1}-{\hat{u}}_{k}^{t}\right\|}_{2}^{2}}{{\left\|{\hat{u}}_{k}^{t}\right\|}_{2}^{2}}<\varepsilon$;Step 2.8:Return $\omega_{k}$, which represent the modes of the original signal.Step3:Screen the 2D mode components of the plurality of BIMFs extracted by 2D-VMD algorithm as feature of human motion.Step4:Reconstruct the original 2D echo signal of human motion according to the 2D mode component of BIMFs.Step5:Evaluate the effect of the 2D-VMD algorithm application in human motion echo signal feature extraction and reconstruction according to the *PCC*, *UQI*, and *PSNR* between the carrier-free UWB radar echo signal of human motion and reconstructed signal.

## 4. Experimental Result

### 4.1. Experimental Setup and Data Acquisition

The 2D human motion echo signal data of the carrier-free UWB radar is acquired by the SIR-20 high-speed ground penetration radar developed by American Lauley Industrial Technology Ltd. (Guilin, China) in this paper. The SIR-20 system is preinstalled with an operating system and acquisition processing software that uses a standard Geophysical Survey Systems (GSSI) antenna. The center frequency of the experimental acquisition is 400 MHz, the bandwidth is 800 MHz, the antenna gain is ~3 dBi, the scanning rate of signal channel acquisition is 100 times/second, and the number of samples pre scan is 512 points. The SIR-20 ground penetrating radar is controlled by a Panasonic PC which has stored the acquired data. The SIR-20 experimental measurement device is shown in Figure 2. Table 1 describes the radar parameters of the SIR-20 device in our experiment.

The experimental measurement is performed indoors with some tables and chairs. A total of ten different types of human motion 2D echo signals were collected, which mainly included (a) walk forward, (b) walk backward, (c) run forward, (d) run backward, (e) fall forward, (f) fall backward, (g) walk around, (h) jump up and down, (i) jump forward, and (j) jump backward.

The SIR-20 device for experimental data acquisition uses a transceiver antenna. The transceiver antenna is placed on a table ~1 m away from the ground. The data acquisition is performed by the lab fellow whose height is about 172 cm and the weight is ~65 kg. The lab fellow faces the radar antenna, which is approximately 2 m away. Each type of motion was repeated 20 times in the data collection, and the duration of data collected by each motion was approximately 120 s. The specific ten different types of human motion description are shown in the Table 2. The measured scenes of the ten different types of human motion (a)~(j) collected in the experiment is shown in Figure 3. The raw data of the experiment are shown in Figure 4.

The experimental data is the acquisition data of ten different types of human motion through SIR-20 high-speed GPR. Figure 5 is a 2D echo signals of ten different types of human motion by carrier-free UWB radar, with horizontal orientation indicating the total distance of the transmitted pulse (fast time) and coordinates indicting sampling time of the transmitted pulse (slow time). In Figure 5, it can be clearly seen that echo signals of different types of human motion have significant differences, which formed by superimposing several different center frequency signals.

The traditional image is composed of several pixels. Compared with the traditional image composed of pixels, the 2D human motion echo signal of the carrier-free UWB radar is composed of several different center frequencies, including the head, torso, hands, legs, etc. Therefore, the original 2D radar echo signal is decomposed into several BIMFs that is 2D mode components by 2D-VMD algorithm. Each BIMFs component represents a certain group center frequency in a certain type of human motion 2D radar echo signal. The set of center frequency signals contains different local frequency components of a certain part of the human motion. Using the 2D-VMD algorithm to extract and reconstruct 2D echo signals of different types of human motion obtained by the carrier-free UWB radar can effectively extract important features in 2D UWB radar echo signal of different types of human motion, which also has a certain inhibitory effect on the clutter from echo signals and the carrier-free UWB radar transmitting device.

### 4.2. Experimental Result

#### 4.2.1. Feature Extraction

In this paper, the 2D-VMD algorithm is used to decompose the 2D echo signals of different types of human motion by carrier-free UWB radar into several 2D BIMFs mode component. Each BIMF mode component represents the different group centers of the decomposed 2D echo signals and represents an important feature of a certain set of center frequencies of human motion [33,34,35]. This article verifies the 2D-VMD algorithm parameter setting is shown in Table 3, the BIMFs component feature extracted by 2D-VMD algorithm of different types of human 2D echo signal is shown in Figure 5, Appendix A, Figure 6, and Appendix A is a reconstructed human motion 2D echo signal.

The 2D echo signal of ten different types of human motion by UWB radar are decomposed by the 2D-VMD algorithm, and several 2D mode components of BIMFs are obtained, the results are shown in Figure 5 and Appendix A. It can be seen from Figure 5 and Appendix A that the original 2D radar echo signal is decomposed by the 2D-VMD algorithm to obtain several 2D BIMFs mode components, which represent different groups of central frequency components of a certain type of human motion. These different groups of central frequency signals contain different local frequency components of a certain part of human motion. The 2D BIMFs mode component represents some unique properties of the original 2D radar echo signal and can extract features of different types of human motion 2D echo signals acquired by the carrier-free UWB radar through 2D-VMD algorithm, which can effectively extract the important features of different types of human motion and has a certain inhibitory effect on the clutter that from echo signals and carrier-free UWB radar transmitting device. Because the form of Wiener filtering is embedded to the 2D-VMD algorithm in the process of iterative updating in the frequency domain, compared with the method based on EMD algorithm, the 2D-VMD algorithm is not susceptible to noise and has a certain inhibitory effect on noise [36,37,38].

It can be seen from Figure 5 and Appendix A that several 2D BIMFs mode components of each type of human motion after 2D-VMD algorithm decomposition represent the corresponding feature of the human motion, wherein the BIMF1 component represents the DC signal, high-frequency mode component and high energy component of the 2D echo signal of the carrier-free UWB radar of the human motion. It can be clearly seen that the BIMF1 mode component mainly describes the overall characteristics of this type of the human motion, and BIMF2–5 represent the low-frequency mode component of the 2D echo signal of human motion acquired by the carrier-free UWB radar in Figure 5 and Appendix A. BIMF2–5 mainly describe the local characteristics of different human motion, indicating the more detailed features of human motion. Most of the energy in 2D echo signal of the human motion is concentrated in the BIMF1 mode component, which describes the overall characteristics of 2D radar echo signal. For example, BIMF1, a type of human motion in Figure 5 and Appendix A, describes the overall feature of the walk forward of the human motion 2D radar echo signal, while BIMF2–5 describe the local feature of the walk forward of the human motion 2D radar echo signal. In particular, it can be seen from the picture of BIMF4 and BIMF5 that the local characteristics of this type of human motion are finer. The 2D mode component of BIMF1–5 reveal some characteristics of 2D radar echo signals of different types of human motion from different parts. In this paper, the effectiveness of the feature extraction of the human motion recognition by carrier-free UWB radar is analyzed from the two coefficients of the Pearson coefficient and the universal image quality index between each BIMFs mode component and the human motion 2D radar echo signal. The specific analysis results are shown in Section 5.

#### 4.2.2. Reconstruction

In this paper, a novel feature extraction method—the 2D-VMD algorithm—is used to conduct mode decompose the different types of human motion. Several 2D BIMF mode components were obtained by the 2D-VMD algorithm, and represent the corresponding feature of this type of human motion. Then, the decomposed BIMFs mode component is reconstructed, which can effectively filter out the noise in the original human motion 2D radar echo signal. There are two important functions for the reconstruction of different types of human motion radar echo signals: (1) it can be proved that the decomposition of human motion echo signal of UWB radar by 2D-VMD algorithm plays a certain role in noise reduction and (2) through reconstructing the radar echo signal, it can be further explained that a number of BIMFs by the 2D-VMD algorithm can well represent the echo signal characteristics of this type of human motion, which is a very effective tool for human motion radar echo signal feature extraction. The four different types of human motion original 2D echo signals and reconstructed 2D echo signals are shown in Figure 6.

Figure 6 and Appendix A show the 2D echo signal and reconstructed 2D echo signal of human motion by the carrier-free UWB radar. It can be seen from Figure 6 and Appendix A that the 2D echo signal of different types of human motion are decomposed by the 2D-VMD algorithm, and then reconstruct original echo signals according to the several 2D BIMF mode components after decomposition. It is found that the reconstructed signal can well represent the original echo signal, and the overall contour of the reconstructed echo signal is basically consistent with the original echo signal. Therefore, it can be further demonstrated by the reconstructed echo signal that it works and is very effective to use 2D-VMD algorithm to extract the characteristics of human motion radar echo signal. it can also suppress the clutter which generated in the echo signal and carrier-free UWB radar transmitting and receiving device.

When the original different types of human motion 2D radar echo signal and reconstruction 2D echo signal of this type of human motion is carefully observed in Figure 6 and Appendix A, found that although the overall contour and local details of the outline of the reconstructed 2D echo signal are basically identical with original, the energy of the reconstructed 2D echo signal is less than the energy of the original 2D echo signal, which may be due to the original 2D signal after 2D-VMD decomposition frequency amplitude of the echo signal of each cluster has been reduced, but there is no damage to the characteristics of the echo signal contains information and it also greatly reduces the noise in the original 2D radar echo signal. The differences between the original 2D radar echo signal and the reconstructed 2D echo signal constituted by several BIMFs mode components after 2D-VMD algorithm is analyzed from the three indicators of *PCC* coefficient, *UQI* coefficient and *PSNR* value. The specific performance analysis results are shown in Section 5.

## 5. Performance Analysis

### 5.1. Pearson Correlation Coefficient

Pearson correlation coefficient (*PCC*), also known as the product difference correlation (or product moment correlation) coefficient, is a method for calculating linear correlation proposed by Pearson in the 20th century. The expression is as follows
(15)$$p_{x,y}=\frac{cor(x,y)}{\delta x\delta y}=\frac{E\left[\left(x-x_{\mu })(y-y_{\mu }\right)\right]}{\delta x\delta y}$$

The *PCC* is obtained by dividing the covariance by the standard deviation of the two variables, although the covariance can reflect the degree of two random variables related (covariance when greater than 0 both positive correlation, when less than 0 indicates both negative correlation), the size of the covariance is not a good measure the relevance of the two random variables. According to the covariance divided by two random variables on the basis of the standard deviation of $\delta^{2}=\frac{\sum_{i=1}^{n}\left(x_{i}-x_{\mu }\right)}{n}$, it is easy to conclude that *PCC* is a value between −1 and 1, and, when the linear relationship of the two variables enhances, the correlation coefficient tends to 1 or −1; when one variable increases and the other variable also increases it indicates that they are positively correlated, and the related the coefficient is greater than 0; if one variable increases, the other variable decreases, which indicates that they are negatively correlated, and the correlation coefficient is less than 0; if the correlation coefficient is equal to 0, there is no linear correlation between them.

### 5.2. Peak Signal Noise Rate

Peak signal noise rate (*PSNR*) is expressed also as follows
(16)$$MSE=\frac{1}{H\times W}\sum_{i=1}^{H}\sum_{j=1}^{W}{\left(X(i,j)-Y(i,j)\right)}^{2}$$
(17)$$PSNR=10\log_{10}\left(\frac{{\left(\max\right)}^{2}}{MSE}\right)$$
where *MSE* represents the mean square error of the current signal X and the reference signal *Y*; H and W are the height and width of the signal, respectively; and Max is the maximum value in 2D analysis signal matrix and the unit dB of the *PSNR*. The larger the value of *PSNR* is, the smaller the distortion will be.

*PSNR* is the most common and widely used signal objective evaluation index, however it is based on the error between corresponding pixel points, that is, based on error-sensitive signal quality evaluation. Since the visual characteristics of the human eye are not taken into consideration (the human eye is highly sensitive to the contrast difference of low spatial frequency, the human eye is more gambling than the difference in brightness contrast, the human eye’s perception of a region is subject to the influence of the surrounding area, etc.), so evaluation results often appear that do not agree with people’s subjective feeling.

### 5.3. Universal Image Quality Index

The Universal Image Quality Index (*UQI*) is a new objective image quality evaluation index proposed by literature [39] in 2002. It is believed that the image distortion is determined by three factors: correlation distortion, brightness distortion, and contrast distortion. Although the indicator does not involve the human vision system, the experiment shows that its effect is significantly higher than the evaluation accuracy of the traditional all-parameter image quality objective evaluation index, root mean square error and peak signal-to-noise ratio. Assuming *X* is the original image and *Y* is the image to be evaluated, the *UQI* expression is
(18)$$UQI=\left(\frac{\sigma_{XY}}{\sigma_{X}\sigma_{Y}}\right)\left(\frac{2\mu_{X}\mu_{Y}}{\mu_{X}^{2}+\mu_{Y}^{2}}\right)\left(\frac{2\sigma_{X}\sigma_{Y}}{\sigma_{X}^{2}+\sigma_{Y}^{2}}\right)$$

In Equation (18), the range is [−1,1]: −1 is the worst effect, while 1 is the best effect. It is considered that the image to be evaluated has no distortion. $\mu_{X},\sigma_{X}^{2}$ are the mean value and variance of original image pixel, respectively; $\mu_{Y},\sigma_{Y}^{2}$ are the mean and the variance of pixel values of the image to be evaluated, respectively; and $\sigma_{XY}$ is the covariance between the original image and the pixel value of the image to be evaluated.

#### 5.3.1. Performance Analysis of Feature Extraction

In this paper, the 2D-VMD algorithm is used to decompose the 2D echo signals of different types of human motion carrier-free UWB radar into several BIMFs 2D mode components. Each BIMF 2D mode component represents some unique characteristics of human motion echo signal. This section through calculating the *PCC* coefficient between the original human motion 2D radar echo signal and each BIMF 2D mode component obtained by the 2D-VMD algorithm, and calculates each BIMF mode component percentage, which reveals every BIMF proportion of the mode characteristics of the component. The higher the percentage of each BIMFs is, the more information features are contained in the component of BIMFs in the original radar echo signal. The specific experimental results are shown in Table 4.

The data in Table 4 reveals the Pearson correlation coefficient (*PCC*) between each BIMFs characteristic mode component and the original 2D echo signal. The larger the *PCC* value, the more information the BIMFs mode component contains in the original 2D echo signal, i.e., the larger the proportion of information in the original 2D echo signal. From Table 4, it is found that the proportion of each BIMFs mode component is different, which represents the information amount of the original 2D echo signal from different parts. The 2D-VMD algorithm is a non-recursive multiscale decomposition method, which can decompose the radar echo signal from high-to-low according to the central frequency. Therefore, BIMF1–5 represent different frequency components in the human motion echo and represent the echo information characteristics of a certain part of human body. Table 4 is visualized as shown in the Figure 7 and displays the proportion of each BIMF mode component more intuitively.

In Figure 7 and Appendix A, it the difference between BIMF mode components and the original 2D echo signals can be seen more clearly. From Section 3, it is known that different BIMF components represent different parts of the original echo signals. BIMF1 mainly represents the overall characteristics of this type of human motion. Therefore, the BIMF1 characteristic mode component accounts for ~40% of the original echo signal. BIMF2–5 represent the low-frequency mode component of this type of human motion, representing the local features of different types of human motion. From Table 4, Figure 7, and Appendix A, at least one mode component of BIMF2–5 is the information containing most of the original 2D echo signal, and the rest may represent some local features. Such as in Table 4a in the type of human action echo signal, which accounts for ~40.54% and BIMF2 and BIMF3 account for 62.96% and 79.1%, respectively, indicating that the most information of the human echo signals, while BIMF4 only accounts for 8.67%. This mode component represents some local features of the original 2D echo signal. From Figure 5 and Appendix A, it can be clearly seen that BIMF4 only contains the local subtleties of this type of human motion.

#### 5.3.2. Performance Analysis of Reconstructed 2D Echo Signal

In this paper, the 2D-VMD algorithm is used to decompose the 2D echo signals of different types of human motion carrier-free UWB radar into several BIMFs 2D mode components. These BIMF mode components are used as the characteristics of this type of human motion, and Section 4.2 already very clearly shows that the analysis of BIMF mode components has validity for this type of human motion, this section uses these BIMF 2D mode components to refact the original 2D echo signal that is analyzed from three indicators, and it is further demonstrated that these BIMFs mode components represent the reliability and effectiveness of the original 2D echo signal. The specific experimental results are shown in Table 5.

The reconstructed 2D echo signal and the original signal are analyzed through the three indicators of *PCC*, *UQI*, and *PSNR* in Table 5 so as to illustrate the effectiveness of using the 2D-VMD algorithm to extract features of human motion 2D echo signals of the carrier-free UWB radar in this paper. The 2D-VMD algorithm is used to decompose 2D echo signals of different types of human motion and the original 2D echo signals are reconstructed by using these decomposed BIMFs 2D mode components. The reconstructed signals are a kind of process of noise reduction derived from the original 2D echo signals, which play a certain filtering role. Because the 2D-VMD algorithm introduces Wiener filtering in the iterative update process, it can decompose original 2D echo signals and reduce noise.

As can be seen from Table 5, this paper not only reconstructs 2D echo signals of ten different types of human motion, but also further explores the optimal number of 2D-VMD decomposition layers. Different 2D-VMD decomposition layers have a great impact on the quality of reconstructed 2D echo signals. According to the literature of the original author [18], when the number of layers of 2D-VMD algorithm is 5 (k = 5), the effect of 2D-VMD algorithm is generally better. Therefore, the number of layers of 2D-VMD decomposition in Section 3 is also five layers, and the results in Section 3 show that the effect is also good. In this section, the difference between the reconstruction of 2D echo signal and the original 2D echo signal is also used to further explore the optimal number using the 2D-VMD algorithm of the human motion 2D echo signal based on the carrier-free UWB radar. k = 2–9 was used to decompose the echo signals of ten different types of human motion 2D carrier-free UWB radar echo signal, respectively, and the effects of different decomposition layers is measured on the reconstructed 2D echo signals by *PCC*, *UQI* and *PSNR*. These experimental results are shown in Table 5, Figure 8 and Appendix A. From Figure 8 and Appendix A it can be seen at a glance that the layers of different types of human motion decomposition are different, and the difference between the performances of the reconstructed 2D echo signals is also relatively large. This section analyzes ten different types of human motion through the 2D-VMD algorithm. The *PCC*, *UQI*, and *PSNR* between the reconstructed 2D echo signal and the original 2D radar echo signal explore the optimal number of layers of the human motion 2D echo signal decomposition of the carrier-free UWB radar. The *PCC* reflects the linear correlation between the reconstructed 2D echo signal and the original 2D radar echo signal. The closer the *PCC* value is to 1, the better the reconstruction performance of the reconstructed 2D echo signal will be. *UQI* is the universal objective evaluation index of the image, and its value is mainly determined by three factors: correlation distortion, brightness distortion, and contrast distortion. Although *UQI* is derived mainly according to the evaluation index of the image, at the same time, it also is closely related to the image luminance distortion and contrast distortion; however, this article reconstructs the 2D radar echo signal, and its signal was essentially a 2D analysis signal that had a huge difference from the image. From Figure 4, Figure 5 and Figure 6, Appendix A it can be found that the 2D analysis signal also be visualized in the form of the image, the intensity of the brightness indicates the energy of the 2D analysis signal, therefore the *UQI* value can also be used to analysis the difference between reconstruction of 2D echo signal and the original 2D radar echo signal. The value range of *UQI* is [−1, 1], and when the value of *UQI* is −1, the effect of reconstructed 2D echo signal is the worst. When the value of *UQI* is 1, the effect of reconstructed 2D signal is the best. *PSNR* is the most common and widely used objective image or signal evaluation index, and it is an error-sensitive image quality evaluation based on *MSE*. The larger the *PSNR* value is, the smaller the distortion of the reconstructed 2D echo signal will be.

From Table 5, Figure 8, and Appendix A, it can be concluded that when k = 5 or k = 6, the reconstructed 2D echo signals can represent the original 2D radar echo signal well. The *PCC*, *UQI*, and *PSNR* between original 2D echo signals and reconstructed 2D echo signals when k = 6 were better than these three indicators when k = 5, only (i) type human motion during decomposition layers to 5 of the *PCC*, *UQI*, and *PSNR* slightly better than decomposition layers is 6. The *PCC*, *UQI*, and *PSNR* of (i) type human motion between reconstructed 2D echo signal and original 2D radar echo signal at k = 5 and k = 6 were 0.9914, 0.86706, and 25.71 and 0.9804, 0.8374, and 21.93, respectively. The decomposition performance of the other nine different types of human motion at k = 6 is better than that at k = 5. Therefore, the optimal number of decomposition layers of 2D-VMD for human motion 2D echo signals carrier-free UWB radar is 6 layers. Ten different types of human motion are compared when k = 5 and k = 6, the 2D signal is used to extract the BIMFs mode components and the reconstructed 2D echo signal using the 2D-VMD algorithm, as shown in Figure 9 and Appendix A.

Figure 9 and Appendix A show the characteristics of BIMF1–6 mode components extracted by 2D-VMD algorithm when k = 6. Compared with Figure 6 and Appendix A, it is found that the BIMF1 mode component is high-frequency mode component and high-energy component of this type of human motion which contains a large amount of information about this motion, expressing the overall characteristics of this type of human motion, but when k = 6, 2D-VMD algorithm decomposition is more thorough, and is able to be more local and has more fine features, like neural networks (CNN) to extract the characteristics of the deeper. Each 2D BIMFs mode component which representing different groups of central frequency components of a certain type of human motion. These different groups of central frequency signals contain different local frequency components on a certain part of human. Therefore, when k = 6, the 2D-VMD algorithm can extract more abundant feature information in human motion radar echo signal, including finer and contour information from different parts of human body. Meanwhile, it is also shown the reconstructed 2D echo signal and the original through the three indicators of *PCC*, *UQI*, and *PSNR* of ten different types of UWB radar with different decompose layers using 2D-VMD algorithm in Table 5. At the same time, it can be seen from Figure 10 that the 2D echo signal reconstructed by the 2D-VMD algorithm can better express the original echo signal when k = 6.

Figure 10 and Appendix A show the contrast between reconstructed 2D echo signals when k = 5 and k = 6; it can be seen that the original 2D radar echo signals are well reconstructed. The overall characteristics of the original echo signal can be reconstructed very well. Therefore, they all can be further demonstrated by the reconstructed echo signal when k = 5 and k = 6. It is very effective to use the 2D-VMD algorithm to extract the characteristics of human motion radar echo signal. On the other hand, it can also suppress the clutter which generated in the echo signal and the carrier-free UWB radar transmitting and receiving device. But when k = 6, the local details of some types of human motion can be expressed more finely, as shown in Figure 10a, when k = 6, the details of some original 2D radar echo signals can be reconstructed more clearly. Therefore, the optimal layer of 2D-VMD decomposition algorithm of the carrier-free UWB radar human motion recognition is 6 layers.

This paper introduces a novel method for feature extraction of human motion echo signal of UWB radar because UWB radar has higher resolution, stronger anti-multipath fading and anti-interference ability than the CW radar. The UWB radar can also penetrate obstacles to identify human, especially in a complex indoor environment. As UWB radar transmits a low-powered pulse, it can provide range information of the target without considering limits of frequency allocation. Furthermore, it has short pulse duration such that it can contain information about a human motion. However, The UWB radar human motion echo signal is a nonlinear, nonstationary signal [40]. Methods such as FFT and STFT are time–frequency analysis for linear and stationary signals. So, the effect of human motion echo signal is relatively poor in this paper. Therefore, a human motion echo signal decomposition method based on the 2D-VMD algorithm to extract the characteristic of this type of human motion echo signal is proposed. The 2D-VMD algorithm is a non-recursive variable-scaled composition iterative search variational model that decomposes the 2D human motion echo signal of the carrier-free UWB radar into several BIMFs mode components, which represent the primary feature of this type of human motion.

The work of this paper mainly focuses on the feature extraction of human motion echo of the carrier-free UWB radar, and based on an adaptive non-recursive multiscale decomposition algorithm, 2D-VMD algorithm, the feature extraction of human motion echo signals of different types is carried out. This is a part of the feature engineering in our research on radar-based human motion recognition. The next step of our research is to use the traditional classification & recognition methods (such as SVM, KNN, decision tree, etc.) and LSTM in deep learning to classify and recognize different types of human motion.

## 6. Conclusions and Future Prospects

A feature extraction method of different types of human motion radar echo signals by using a novel 2D-VMD algorithm is studied and discussed in this paper. The 2D-VMD algorithm is a non-recursive variable-scaled decomposition method for nonlinear and nonstationary signals. We use 2D-VMD algorithm to decompose a human motion radar echo signal into a discrete number of subsignals (modes), where each mode can represent the primary feature of human motion radar echo signal. Then, the BIMFs are utilized to reconstruct original radar echo signal, which not only suppressed the clutter and noise in the echo signals by reconstructing the original echo signals, but also further illustrate that the BIMFs modal components obtained by the 2D-VMD decomposition algorithm can well represent the original 2D echo signals.

In our paper, ten different types of human motion UWB radar 2D echo analysis signals are tested and verified experimentally. The experimental results show that the original 2D echo signal map of UWB radar is decomposed by using the 2D-VMD algorithm, and several BIMF mode components representing the local frequency components of different groups of center frequencies of a certain type of human motion are obtained. The 2D mode components of BIMFs reveal some characteristics of different types of human motion 2D radar signals from different parts, among which, BIMF1 represents the overall characteristics of this type of human motion echo signal, and BIMF2–5 represent the local fine characteristics of this type of human motion echo signal. Secondly, the BIMF1 component indicate the DC signal, high-frequency mode component and high energy component of human motion echo signal of the carrier-free UWB radar, and the BIMF2–5 components indicate the low-frequency component. Thirdly, according to several BIMF mode components by 2D-VMD decomposition algorithm reconstructed by the original 2D echo signal, after 2D-VMD decomposition of each variety of different types of human motion the echo signal frequency amplitude reduced. However, the characteristics information contained in the echo signal is not substantially damaged, and the noise in the original 2D radar echo signal is also greatly reduced. This paper also quantitatively analyzes the feature extraction and reconstruction of the original echo signals from the three indicators of the *PCC*, *UQI*, and *PSNR*, so as to illustrate the effectiveness of using 2D-VMD algorithm to extract features of the human motion 2D echo signals of the carrier-free UWB radar. Finally, we further explored the effect of different decomposition layers from k = 2 ~ k = 9, and found that the 2D echo signal reconstructed by the 2D-VMD algorithm can better express the original echo signal when k = 6.

The experimental results of this paper are based on the measured data in MATLAB platform. Since the 2D human motion echo signal of UWB radar is different from the traditional pixel-based image, a novel 2D-VMD signal decomposition algorithm is proposed to extract and reconstruct the features of different types of human motion radar echo signals. In this paper, the 2D-VMD algorithm is used to decompose the UWB radar echo signals mainly in order to extract the key features of different types of human motion echo signals, and to better distinguish different types of human motion by comparing the differences in features between different types of human motion, which is the next step of our research plan. The next step of our research is to use the traditional classification & recognition methods (such as SVM, KNN, decision tree, etc.,) and LSTM in deep learning to classify and recognize different types of human motion. The feature extraction method by 2D-VMD algorithm is also useful for reference for further research on multidimensional analysis signal decomposition and radar-based human motion recognition.

## Figures and Tables

**Figure 1 sensors-19-01962-f001:**
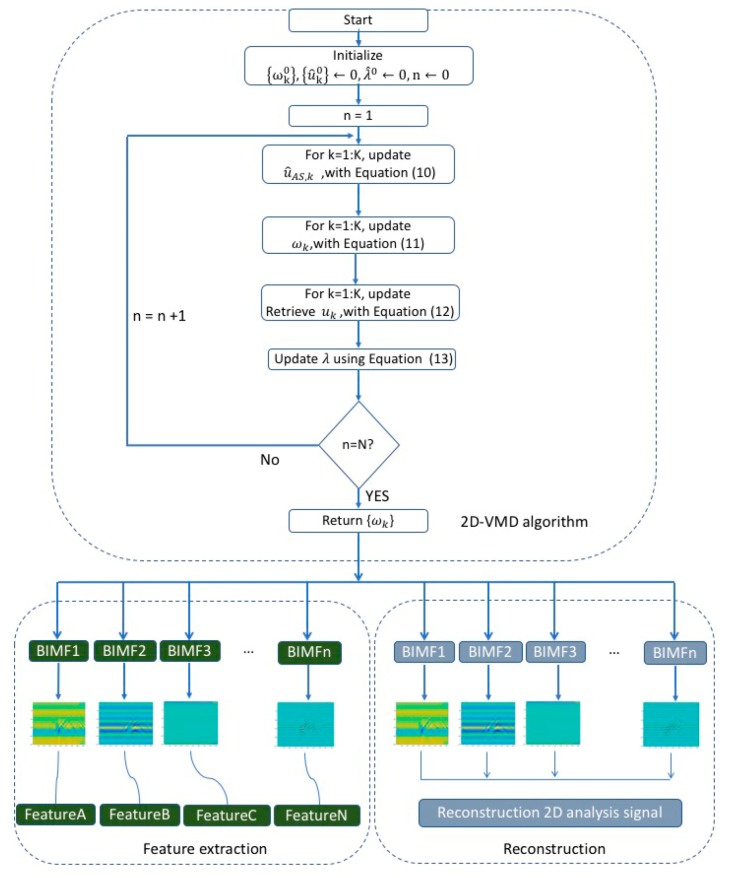
The schematic diagram of feature extraction and reconstruction model framework of human motion 2D echo signal based on two-dimensional variational mode decomposition (2D-VMD) algorithm by carrier-free ultra-wideband (UWB) radar in our paper.

**Figure 2 sensors-19-01962-f002:**
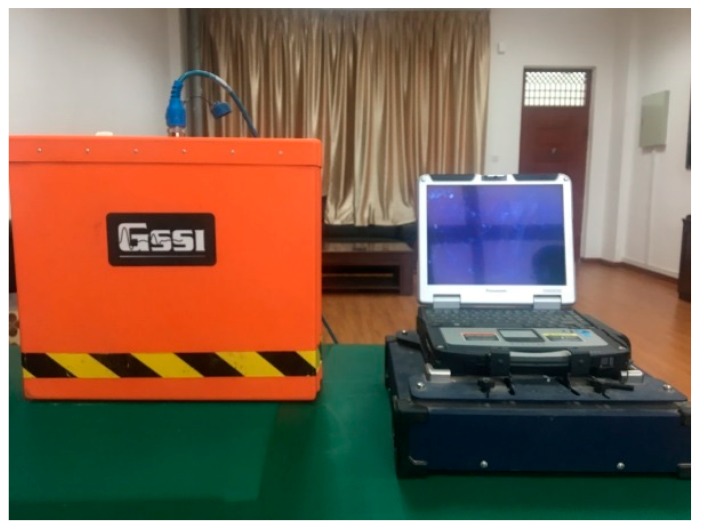
SIR-20 ground penetrating radar.

**Figure 3 sensors-19-01962-f003:**
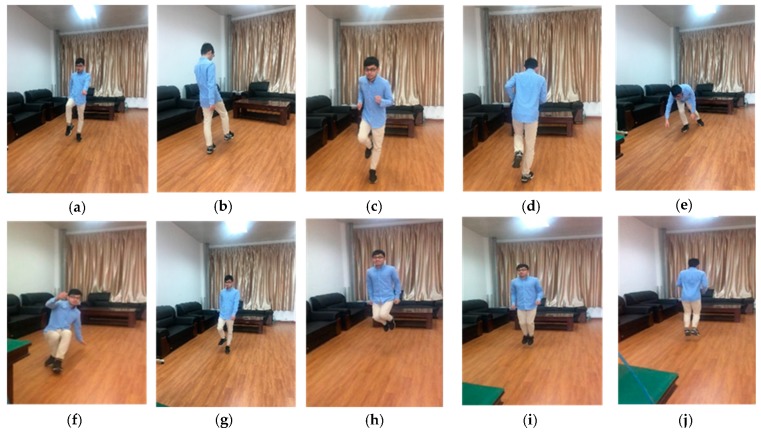
Ten different types of human motion measurement scenes, (**a**) walk forward of human motion measurement scenes, (**b**) walk backward of human motion measurement scenes, (**c**) run forward of human motion measurement scenes, (**d**) run backward of human motion measurement scenes, (**e**) fall forward of human motion measurement scenes, (**f**) fall backward of human motion measurement scenes, (**g**) walk around of human motion measurement scenes, (**h**) jump up and down of human motion measurement scenes, (**i**) jump forward of human motion measurement scene, (**j**) jump backward of human motion measurement scenes.

**Figure 4 sensors-19-01962-f004:**
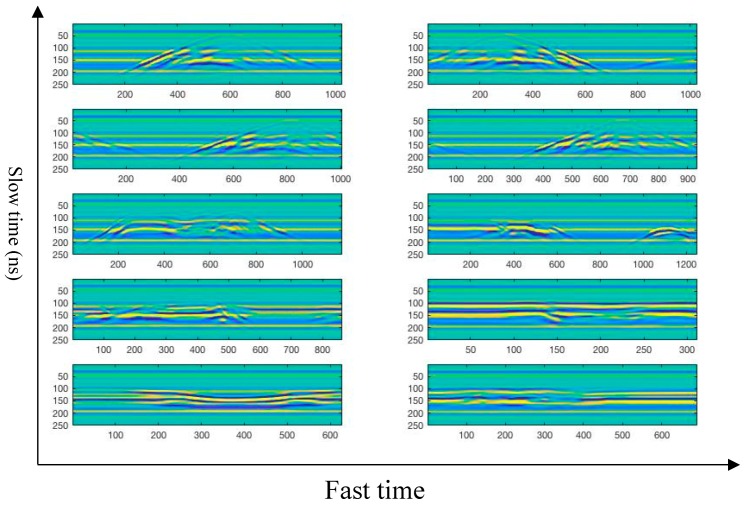
2D echo signal map of ten different types of human motion. (from left to right, from top to bottom, motion a ~ motion j 2D echo signal).

**Figure 5 sensors-19-01962-f005:**
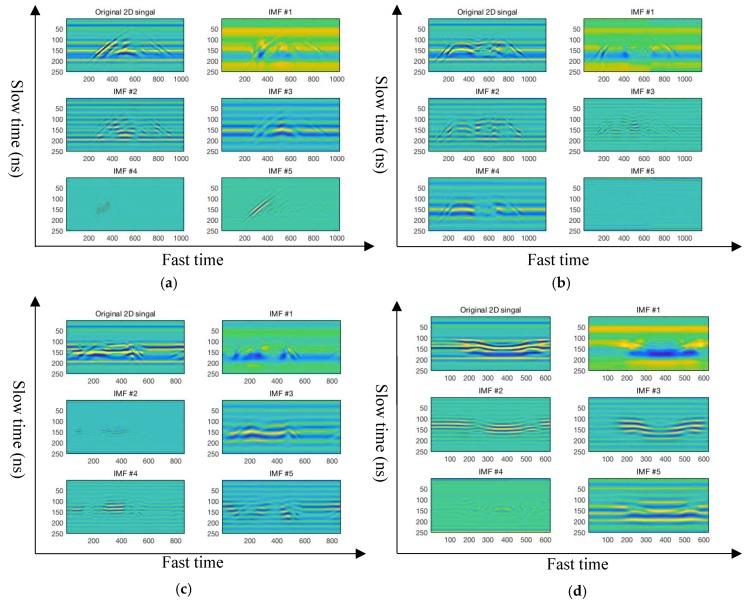
Four different types of human motion 2D echo signals and several BIMF components after 2D-VMD decomposition. (**a**) Walk forward of human motion 2D echo signal and several BIMF components; (**b**) fall forward of human motion 2D echo signal and several BIMF components; (**c**) walk around of human motion 2D echo signal and several BIMF components; and (**d**) jump forward of human motion 2D echo signal and several BIMF components.

**Figure 6 sensors-19-01962-f006:**
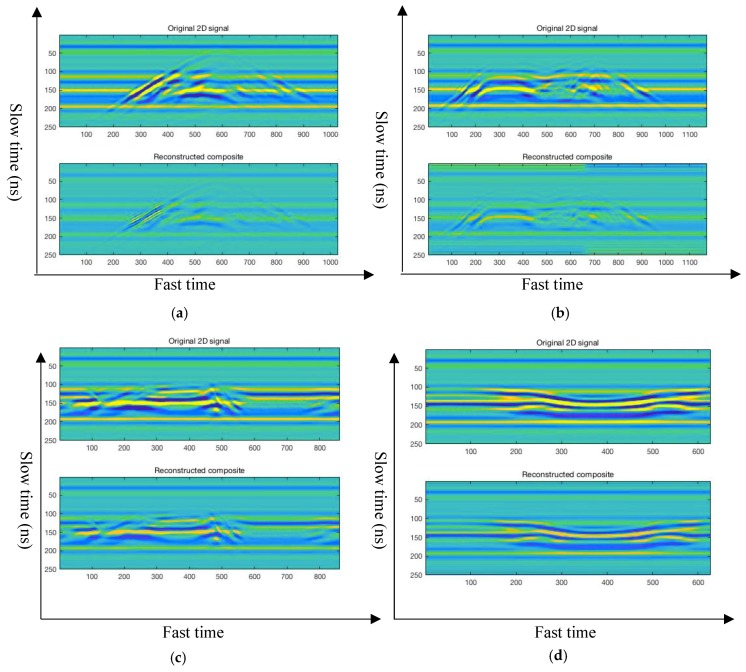
Four different types of human motion original 2D echo signals and reconstructed 2D echo signals. (The above figure shows the original human motion radar 2D echo signal, the blow figure shows the reconstructed 2D radar echo signal.) (**a**) Walk forward of human motion original 2D echo signal and reconstructed 2D echo signal; (**b**) fall forward of human motion original 2D echo signal and reconstructed 2D echo signal; (**c**) walk around of human motion original 2D echo signal and reconstructed 2D echo signal; and (**d**) jump forward of human motion original 2D echo signal and reconstructed 2D echo signal.

**Figure 7 sensors-19-01962-f007:**
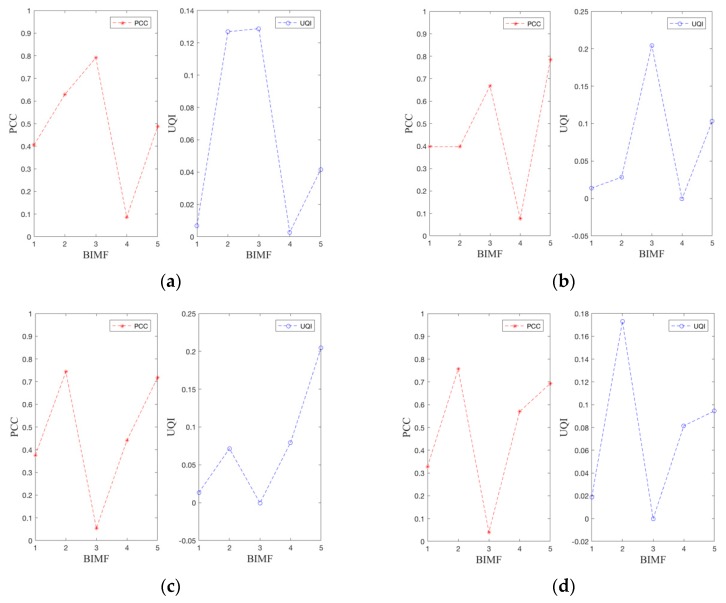
*PCC* and *UQI* indicators for feature extraction of human motion 2D echo analysis signals of UWB radar. (**a**) The *PCC* and *UQI* indicators for feature extraction of walk forward of human motion 2D echo analysis signal. (**b**) The *PCC* and *UQI* indicators for feature extraction of fall forward of human motion 2D echo analysis signal. (**c**) The *PCC* and *UQI* indicators for feature extraction of walk around of human motion 2D echo analysis signal. (**d**) The *PCC* and *UQI* indicators for feature extraction of jump forward of human motion 2D echo analysis signal.

**Figure 8 sensors-19-01962-f008:**
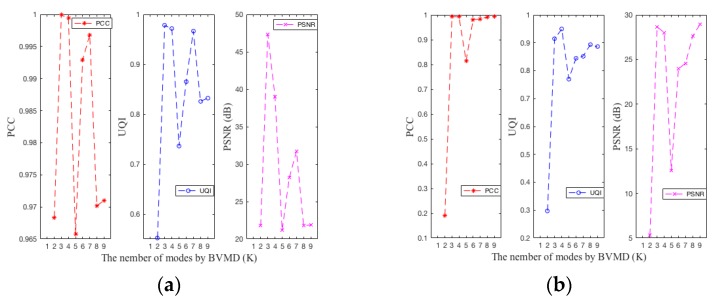
Three indicators of *PCC*, *UQI*, and *PSNR* of 2D echo signal reconstruction of four different types of UWB radar-based human motion 2D echo analysis signal. (**a**) The three indicators of *PCC*, *UQI*, and *PSNR* of 2D echo signal reconstruction of walk forward of UWB radar-based human motion 2D echo analysis signal. (**b**) The three indicators of *PCC*, *UQI*, and *PSNR* of 2D echo signal reconstruction of fall forward of UWB radar-based human motion 2D echo analysis signal. (**c**) The three indicators of *PCC*, *UQI*, and *PSNR* of 2D echo signal reconstruction of walk around of UWB radar-based human motion 2D echo analysis signal. (**d**) The three indicators of *PCC*, *UQI*, and *PSNR* of 2D echo signal reconstruction of jump forward of UWB radar-based human motion 2D echo analysis signal.

**Figure 9 sensors-19-01962-f009:**
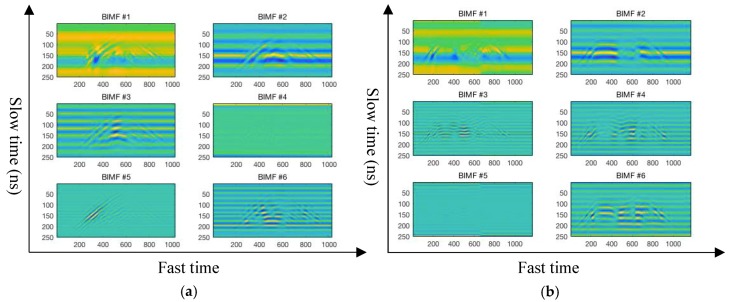
The six BIMF mode components of feature extraction of the carrier-free UWB radar human motion 2D echo analysis signal when the 2D-VMD decomposition layer number is six. (**a**) The six BIMF mode components of feature extraction of walk forward of the carrier-free UWB radar human motion 2D echo analysis signal. (**b**) The six BIMF mode components of feature extraction of fall forward of the carrier-free UWB radar human motion 2D echo analysis signal. (**c**) The six BIMF mode components of feature extraction of walk around of the carrier-free UWB radar human motion 2D echo analysis signal. (**d**) The six BIMF mode components of feature extraction of the carrier-free UWB radar human motion 2D echo analysis signal.

**Figure 10 sensors-19-01962-f010:**
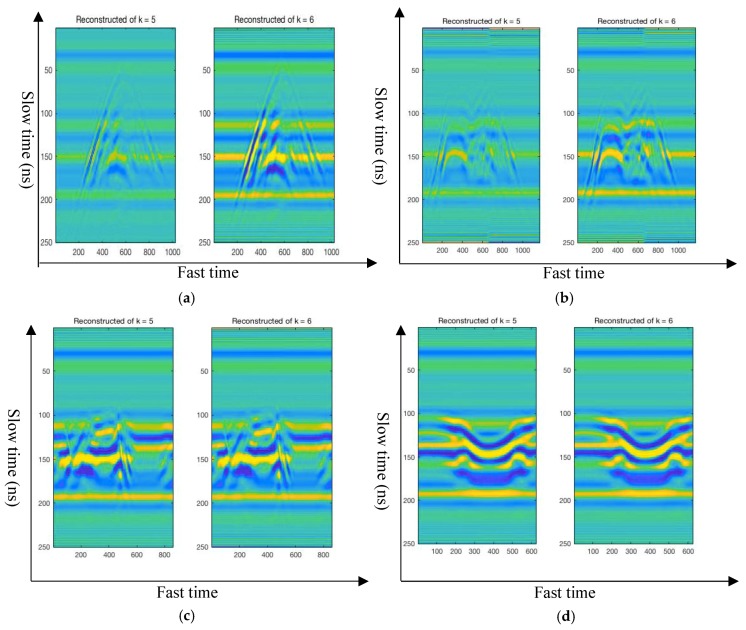
Reconstruction of 2D human motion echo analysis signals of four different types of carrier-free UWB radar. (The figure on the left shows the reconstruction of 2D echo signal of human motion of the carrier-free UWB radar through 2D-VMD algorithm when k = 5, and the figure on the right shows the reconstruction of 2D echo signal of human motion of carrier-free UWB radar through 2D-VMD algorithm when k = 6.) (**a**) Reconstruction of 2D human motion echo analysis signal of walk forward of carrier-free UWB radar. (**b**) Reconstruction of 2D human motion echo analysis signal of fall forward of carrier-free UWB radar. (**c**) Reconstruction of 2D human motion echo analysis of walk around of carrier-free UWB radar. (**d**) Reconstruction of 2D human motion echo analysis signal.

**Table 1 sensors-19-01962-t001:** Experimental radar parameter table.

Parameter	Value
Center frequency	400 MHz
Time window	20 ns
Scanned sample points	512 p/s
Resolution	16 Bit
Scanning frequency	50 Hz
Transmit repetition rate	100 K

**Table 2 sensors-19-01962-t002:** Specific ten different types of human motion description.

Number	Motion Category	Specific Motion Description
(a)	Walk forward	The two hands swing alternately and walk slowly towards the antenna towards the radar
(b)	Walk backward	The two hands swing alternately, starting close to the antenna position, and slowly walk backward, slowly moving away from the antenna
(c)	Run forward	The two hands swing alternately and run toward the radar antenna
(d)	Run backward	The two hands swing alternately, starting close to the antenna and run backward, away from the antenna
(e)	Fall forward	Standing 2 m away from the antenna, fall forward slowly, and finally lying on the ground
(f)	Fall backward	Standing 2 m away from the antenna, fall backward slowly, and finally lying on the ground
(g)	Walk around	Standing 2 m away from the antenna and walk around
(h)	Jump up and down	Standing 2 m away from the antenna, up and down in a periodic beat
(i)	Jump forward	Standing 2 m away from the antenna, continuously jumping forward at a constant speed
(j)	Jump backward	Starting point close to the antenna position, continuously jumping backward at a constant speed

**Table 3 sensors-19-01962-t003:** 2D-VMD algorithm parameter in the paper.

Parameters	Meaning	Value
alpha	the balancing parameter for data fidelity constraint	5000
tau	time-step of dual ascent (pick 0 for noise-slack)	0.25
K	the number of IMFs to be recovered	5
DC	true, if the first mode is put and kept at DC (0-freq)	1
init	0, if all omegas start at 0 1, if all omegas start initialized randomly	1
tol	tolerance of convergence criterion	$5\times 10^{-6}$

**Table 4 sensors-19-01962-t004:** The number and percentage of *PCC* about 2D signal and BIMF.

	BIMF1	BIMF2	BIMF3	BIMF4	BIMF5	Max
**(a) type carrier-free UWB radar human motion 2D echo signal**						
*PCC*	104,080	161,640	203,096	22,260	125,302	256,749
	40.54%	62.96%	79.1%	8.67%	48.8%	---
**(b) type carrier-free UWB radar human motion 2D echo signal**						
*PCC*	108,615	174,474	16,994	202,551	119,377	256,749
	42.3%	67.96%	6.62%	78.89%	46.5%	---
**(c) type carrier-free UWB radar human motion 2D echo signal**						
*PCC*	85,874	102,506	153,095	178,683	154,012	251,249
	34.18%	40.8%	60.93%	71.12%	61.3%	---
**(d) type carrier-free UWB radar human motion 2D echo signal**						
*PCC*	99,881	159,628	184,590	77,948	17,432	233,249
	42.82%	68.47%	79.14%	33.42%	7.47%	---
**(e) type carrier-free UWB radar human motion 2D echo signal**						
*PCC*	116,402	116,483	196,009	22,401	230,051	293,249
	39.69%	39.72%	66.84%	7.64%	78.45%	---
**(f) type carrier-free UWB radar human motion 2D echo signal**						
*PCC*	112,246	219,052	126,029	231,726	17,090	312,499
	35.92%	70.1%	40.33%	74.15%	5.47%	---
**(g) type carrier-free UWB radar human motion 2D echo signal**						
*PCC*	80,762	160,198	11,623	95,162	154,590	215,499
	37.48%	74.34%	5.39%	44.16%	71.74%	---
**(h) type carrier-free UWB radar human motion 2D echo signal**						
*PCC*	26,662	30,706	52,388	56,167	38,701	77,749
	34.29%	39.49%	67.38%	72.24%	49.78%	---
**(i) type carrier-free UWB radar human motion 2D echo signal**						
*PCC*	51,368	118,507	6248	89,280	108,569	156,749
	32.77%	75.6%	3.99%	56.96%	69.26%	---
**(j) type carrier-free UWB radar human motion 2D echo signal**						
*PCC*	53,625	78,197	97,097	24,057	138,874	172,749
	31.04%	45.27%	56.21%	13.93%	80.39%	---

**Table 5 sensors-19-01962-t005:** Three indicators of *PCC*, *UQI*, and *PSNR* for 2D echo signal reconstruction of ten different types of UWB radar with different decomposition layers in 2D-VMD algorithm.

	K = 2	K = 3	K = 4	K = 5	K = 6	K = 7	K = 8	K = 9	Max Value
**(a) type carrier-free UWB radar human motion 2D echo signal**									
*PCC*	248,617	256,726	256,597	247,953	254,928	255,932	249,085	249,307	256,749
	96.83%	99.99%	99.94%	96.57%	99.29%	99.68%	97.02%	97.10%	
*UQI*	0.55205	0.97805	0.9716	0.73616	0.86540	0.96673	0.82585	0.83218	1
*PSNR* (dB)	21.77	47.33	39.03	21.15	28.18	31.72	21.76	21.89	---
**(b) type carrier-free UWB radar human motion 2D echo signal**									
*PCC*	247,787	256,059	241,372	253,784	254,555	248,860	241,441	254,916	256,749
	96.51%	99.73%	94.01%	98.85%	99.15%	96.93%	94.04%	99.29%	
*UQI*	0.4776	0.97532	0.81668	0.93662	0.94284	0.83562	0.81843	0.87802	1
*PSNR* (dB)	19.51	32.16	18.20	25.84	27.16	21.35	18.22	27.92	---
**(c) type carrier-free UWB radar human motion 2D echo signal**									
*PCC*	250,961	251,246	251,197	251,117	251,249	251,249	251,248	251,249	251,249
	99.89%	99.99%	99.98%	99.95%	100%	100%	99.99%	100%	
*UQI*	0.88342	0.98640	0.95510	0.96638	0.98709	0.98738	0.98531	0.98791	1
*PSNR* (dB)	36.35	56.28	43.78	39.72	64.80	66.33	59.44	67.70	---
**(d) type carrier-free UWB radar human motion 2D echo signal**									
*PCC*	228,903	194,715	220,596	201,151	221,035	223,169	230,969	227,036	233,249
	98.14%	83.48%	94.58%	86.24%	94.76%	99.96%	99.02%	97.34%	
*UQI*	0.60115	0.77158	0.82678	0.79566	0.80585	0.80606	0.90814	0.81756	1
*PSNR* (dB)	24.08	12.28	19.06	14.27	19.20	20.11	26.91	22.39	---
**(e) type carrier-free UWB radar human motion 2D echo signal**									
*PCC*	56,054	291,277	290,966	238,999	287,545	288,283	290,736	291,399	293,249
	19.11%	99.33%	99.22%	81.50%	98.05%	98.31%	99.14%	99.37%	
*UQI*	0.29763	0.91554	0.94855	0.77029	0.84442	0.85204	0.89245	0.88551	1
*PSNR* (dB)	5.34	28.66	28.02	12.59	23.96	24.58	27.61	28.91	---
**(f) type carrier-free UWB radar human motion 2D echo signal**									
*PCC*	303,796	291,571	240,028	299,649	309,522	306,990	307,261	308,575	312,499
	97.22%	93.30%	76.81%	95.89%	99.05%	98.24%	98.32%	98.74%	
*UQI*	0.49477	0.70406	0.74781	0.82720	0.86639	0.8460	0.80333	0.84758	1
*PSNR* (dB)	20.72	18.02	10.77	20.01	26.63	23.87	24.09	25.35	---
**(g) type carrier-free UWB radar human motion 2D echo signal**									
*PCC*	209,543	184,028	215,335	211,657	212,841	210,810	213,809	208,729	215,499
	97.24%	86.39%	99.92%	98.22%	98.77%	97.82%	99.22%	96.86%	
*UQI*	0.56117	0.51705	0.97469	0.84398	0.85368	0.83513	0.86004	0.83124	1
*PSNR* (dB)	21.10	13.27	37.27	23.47	25.14	22.58	27.11	20.89	---
**(h) type carrier-free UWB radar human motion 2D echo signal**									
*PCC*	75,522	77,747	77,748	77,749	77,749	77,749	77,749	77,660	77,749
	97.14%	99.99%	99.99%	100%	100%	100%	100%	99.89%	
*UQI*	0.52529	0.97924	0.97985	0.98051	0.98103	0.98140	0.98167	0.90281	1
*PSNR* (dB)	18.73	51.27	54.77	57.45947	59.42	60.59	61.23	33.85	---
**(i) type carrier-free UWB radar human motion 2D echo signal**									
*PCC*	154,341	156,737	152,633	155,394	153,670	155,549	156,155	156,292	156,749
	98.46%	99.99%	97.37%	99.14%	98.04%	99.23%	99.62%	99.71%	
*UQI*	0.60568	0.98305	0.83415	0.86706	0.83474	0.86286	0.91473	0.89895	1
*PSNR* (dB)	23.28	46.31	20.57	25.71	21.93	26.25	29.33	30.44	---
**(j) type carrier-free UWB radar human motion 2D echo signal**									
*PCC*	145,729	172,746	172,748	170,687	171,190	172,003	172,394	172,609	172,749
	84.36%	99.99%	99.99%	98.81%	99.09%	99.57%	99.79%	99.92%	
*UQI*	0.20419	0.98297	0.98348	0.85448	0.86438	0.87910	0.90366	0.95810	1
*PSNR* (dB)	13.69	53.07	56.98	24.79	26.03	29.22	32.48	36.52	---

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
