# Peer review of "Feature Extraction and Reconstruction by Using 2D-VMD Based on Carrier-Free UWB Radar Application in Human Motion Recognition"

_sensors, 2019, doi:10.3390/s19091962_

Round 1
Reviewer 1 Report
Dear Authors
The paper has some interesting reading but overall it is too busy . Technical details when they are needed are not present.
e.g a) the overhead the work ensues to provide some image analysis against existing recognition methods, b) processing time of the work alone, c) adaptively to other moves, you mention classification but I cannot see how you do this with the current method.
I think it would be helpful to 1) clarify what you are trying to do and make your abstract clearer and more pointed. 2) Reduce some of the more abstract information an employ the mathematical methods that are actually directly related to the work, thir function and using your tests to explore their context .i.e. explain the method and how it effects the results. Some of your Figure to Appendix comparisons are not useful. Also, figures etc. should all be labelled.
Overall the paper is longwinded, it could be maintained in length but needs more discussion on the method and the results section. Readers are familiar with the theory, what they would like to read is why use are using those methods over others, and a comparison on why your method is an improvement (if in fact it is?). Throwing lots of theory in at the beginning, unless you add context to your work leads to wondering why it’s there. It should not be there to fill pages- use the vital space to highlight the method, show how theort supports it and how it fairs with other methods, emphasising the new work.
The paper needs to be reviewed for grammatical and presentation errors.
Author Response
Dear Editors and Reviewers:
We are very sorry for our incorrect writing to give you reading difficulties. This is my first English paper, so there are a lot of problems in the paper. Thank you for your letter and for the reviewer's comments concerning our manuscript entitled "Feature extraction and reconstruction by using 2D-VMD based on carrier-free UWB radar application in human motion recognition" (ID: sensors-462644). Those comments are all valuable and very helpful for revising and improving our paper, as well as the important guiding significance to our researches. We have studied comments carefully and have made correction which we hope meet with approval. Revised portion are marked in the paper. The main corrections in the paper and the responds to the reviewer's comments in the attached Word file.
We tried our best to improve the manuscript and made some changes in the manuscript. These changes will not influence the content and framework of the paper. And here we did not list the changes but marked in revised paper.
We appreciate for Editors/Reviewer’s warm work earnestly, and hope that the correction will meet with approval.
Once again, thank you very much for your comments and suggestions.
Sincerely yours,
Mr. Xiaolong Zhou (Jeff Zhou)

Reviewer 2 Report
This is a lengthy paper with a lot of equations coming from existing works. On the other hand, the novelty of this work is not very clear. I would like to suggest to make this paper concise and focus on the part that is new. In addition, there exist several other issues in this paper. The details are listed below:
The authors are making some absolute statements without providing sufficient proofs. For example, "Compared with other traditional continuous wave radars, the human motion echo signal of the carrier-free UWB radar contains more abundant human motion characteristics information...". Which kind of information can only be obtained by a UWB radar instead of a continuous radar?
I don't understand why time-frequency analysis is not applicable for UWB radar. There are quite a lot papers for micro-Doppler analysis with UWB radar systems?
Another strong statement is in page 2, "The traditional time-frequency analysis method for extracting micro-Doppler features is not suitable for feature extraction...". Please provide proofs.
The authors claim EMD-based methods have some apparent limitations, and VMD algorithm is better. However, the authors didn't provide a clear comparison that how VMD algorithm addresses the issues of EMD-based methods.
The performance analysis part doesn't make any sense, since there is no comparison or guidelines for PCC, PSNR and UQI.
There are many grammar errors, which make this paper difficult to read.
Author Response

(The authors gave the same response as above.)

Round 2
Reviewer 2 Report
Thank you for answering my comments. However, I am still not agree with some statements in the revised version.
In abstract, the authors claim "Compared with other traditional continuous wave radar, the human motion echo signal of the carrier-free UWB radar contains more abundant characteristics information of human motion". What kind of characteristics that only can be obtained through an UWB radar instead of a CW radar?
What do you mean "nonlinear and non-stationary signals"? Why are a carrier free UWB signals nonlinear and non-stationary?
Author Response
Dear Editors and Reviewers:
Thank you for your letter and for the reviewer's comments concerning our manuscript entitled "Feature extraction and reconstruction by using 2D-VMD based on carrier-free UWB radar application in human motion recognition" (ID: sensors-462644). Those comments are all valuable and very helpful for revising and improving our paper, as well as the important guiding significance to our researches. We have studied comments carefully and have made correction, which we hope to meet with approval. Revised portion are marked in the paper. I’m very sorry that some descriptions in our paper may have been ambiguous due to the cultural differences, and we have made corresponding modifications. The main corrections in the paper and the responds to the reviewer's comments in the attached Word file.
We tried our best to improve the manuscript and made some changes in the manuscript. These changes will not influence the content and framework of the paper. And here we did not list the changes but marked in revised paper.
We appreciate for Editors/Reviewer’s warm work earnestly, and hope that the correction will meet with approval.
Once again, thank you very much for your comments and suggestions.
Sincerely yours,
Mr. Xiaolong Zhou (Jeff Zhou)

Round 3
Reviewer 2 Report
As it has been claimed by the authors that the echo signals of any radar systems are nonlinear and non-stationary. FFT and STFT have been widely used in human motion analysis for these nonlinear and non-stationary radar signals. However, the authors claim "The methods such as FFT, STFT are time-frequency analysis for linear and stationary signals". These claims are conflict with each other.
Author Response

(The authors gave the same response as above.)
